# Autonomous Driving System Architecture with Integrated ROS2 and Adaptive AUTOSAR

**Dongwon Hong and Changjoo Moon \*** 

Department of Smart Vehicle Engineering, Konkuk University, Seoul 05029, Republic of Korea;
dks01972@konkuk.ac.kr
\* Correspondence: cjmoon@konkuk.ac.kr

**Abstract:** In the automotive industry, research is now underway to apply Adaptive Automotive Open System Architecture (AUTOSAR) to the development of next-generation mobility, such as autonomous driving and connected cars. However, research on autonomous driving is being predominantly conducted on the robotics platform ROS2 (Robot Operating System 2). This demonstrates a considerable distance between autonomous driving research and its application in actual vehicles. To bridge this gap, interoperability that leverages the strengths of the Adaptive AUTOSAR and ROS2 platforms and compensates for their weaknesses is required. Therefore, this study proposes an architecture for interoperability between the two platforms, named Autonomous Driving System with Integrated ROS2 and Adaptive AUTOSAR (ASIRA). The proposed architecture enables communication between each of the two platforms through the ROS2 SOME/IP Bridge and allows for the necessary data exchange. It validates them in autonomous driving scenarios and goes beyond vehicle development, testing, and prototyping to exploit the advantages of each platform. Additionally, the simulation of autonomous vehicles within the ASIRA architecture is demonstrated by interoperating the ROS2 representative open-source autonomous driving project, Autoware, with the Adaptive AUTOSAR simulator. This study contributes to the assimilation of ROS2 into the automotive industry and its application in real vehicles by linking ROS2 and Adaptive AUTOSAR.

**Keywords:** autonomous vehicles; Adaptive AUTOSAR; ROS2; interoperability; SOME/IP

## 1. Introduction

As the automotive industry becomes increasingly sophisticated, the need for a greater number of sensors and electronic devices has led to increased complexity in the internal configurations of vehicles [1]. At the same time, vehicles must ensure safety, stability, security, as well as the real-time transmission and reception of data. Not only has the number of Electronic Control Units (ECUs) within vehicles grown, but the complexity of the required software has also exponentially increased. Modern vehicle architectures are challenging to manage and may connect over 100 ECUs [2]. This signifies the need for a transformation in the traditional electrical and electronic (E/E) architecture of vehicles.

In response to these advancements, the automotive industry launched "Automotive Open System Architecture (AUTOSAR)", which is an international consortium consisting of automotive OEMs, suppliers, and other industry stakeholders [3]. Since 2003, AUTOSAR has introduced the Classic Platform, a solution for embedded systems with limited resources that offers a high level of safety and meets the Automotive Safety Integrity Level (ASIL-D). AUTOSAR is based on signal-based communication, such as Controller Area Network (CAN) or FlexRay. However, as highly advanced technologies such as autonomous driving and connected cars are developed, and high-performance sensors that continuously produce large volumes of data are incorporated into vehicles, traditional communications such as CAN are beginning to demonstrate bandwidth problems. Furthermore, the flexibility of architectures in making sensor data available to various software applications has

become increasingly critical, and high-performance processors for the smooth acquisition and processing of sensor data have become more important.

To meet these technological demands, AUTOSAR introduced Adaptive AUTOSAR in 2017, which is based on POSIX OS [4]. Adaptive AUTOSAR supports the high-bandwidth Ethernet-based Scalable Service-Oriented Middleware on Ethernet (SOME)/IP protocol and enables a shift from traditional signal-based data communication to service-based communication. Additionally, the support of High-Performance Computing has made it easier to use the computing performance necessary for autonomous driving and to facilitate the integration of various high-performance sensors and algorithms. This advancement is aimed at the development of next-generation mobility, such as autonomous driving. The automotive industry alongside AUTOSAR has already achieved substantial progress in this direction [5].

In contrast to the efforts within the automotive industry, most research and development for autonomous driving has been conducted on the robotics platform known as Robot Operating System (ROS), which is managed and developed by Open Robotics [6]. Initially designed as middleware for research purposes within universities and research institutions, ROS was made available as an open-source project, which allowed for a large number of users to develop and distribute a wide variety of packages and libraries. This openness remarkably accelerated the development speed by making available numerous drivers that supported the sensors necessary for autonomous driving, as well as sensor processing algorithms. Moreover, ROS supported powerful simulation environments such as Gazebo [7] and visualization tools such as Rviz [8], which added to the convenience of development. However, ROS lacked real-time control capabilities and required high computing power. It also used a proprietary communication method, TCPROS [9], which relied on a single point of failure, the ROS Master [10]. TCPROS was unsuitable for industrial use because of the severe security risks associated with exposing the Master IP and Port. To address these issues, a second version, ROS2 [11], was developed.

ROS2 addresses the limitations associated with the research-focused nature of ROS1 by incorporating the Data Distribution Service (DDS) [12], which is a standard used in the military and aviation industries for Service-Oriented Communication (SOC). Additionally, the introduction of DDS Real-Time Publish Subscribe (DDS-RTPS) enables real-time control under the assumption of well-structured code. These innovations demonstrate the potential of ROS1 to move beyond its limitations and progress industrially. For instance, Apex successfully developed a high-safety solution that met the requirements of the ASIL-D level of the functional safety standard for electrical and/or electronic systems (ISO 26262) based on ROS2 [13,14]. This trend has led various ROS1 autonomous driving projects to transition to ROS2; Autoware [15] is a prominent example of this trend. Autoware is an open-source project that encompasses essential autonomous driving functionalities such as perception, decision making, and control [16]. It is currently being tested on various testbeds, including autonomous buses, shuttles, and Autonomous Valet Parking, with ongoing development toward the activation and commercialization of autonomous driving technologies. However, for consumers to use autonomous vehicles in daily life, stringent conditions must be met. Almost all vehicle companies currently adhere to the AUTOSAR standard. This indicates that there is a notable gap between the research and development of autonomous driving technologies and their application in actual vehicles. To leverage the strengths of each platform and mitigate its weaknesses, interoperability between Adaptive AUTOSAR and ROS2 platforms must be ensured in a form that can be realistically applied in the automotive industry.

Two data communication methods exist to ensure such interoperability: DDS and SOME/IP. While Adaptive AUTOSAR is being actively researched and applied, it can lead to unpredictable behavior due to non-determinism issues [17]. Therefore, for stability in the actual vehicle development and production stages, a combined architecture with Classic AUTOSAR, which can achieve deterministic execution, is utilized. Classic AUTOSAR supports SOME/IP but not DDS. Additionally, from a cost perspective, the semiconductor

chips used in Classic AUTOSAR possess very limited hardware resources. DDS covers a significantly broader range of protocols and, due to its various Quality of Service (QoS) features, demands much more memory than SOME/IP. Consequently, compared to SOME/IP, DDS relies heavily on the hardware resources of the vehicle's network infrastructure, and implementing and using DDS on microcontrollers is highly limited in functional aspects.

For these reasons, we propose a method for integrating Adaptive AUTOSAR and ROS2 via the Ethernet-based SOME/IP protocol to ensure safety while maintaining a flexible environment for the development and testing of autonomous vehicles. Adaptive AUTOSAR, which has long been a standard architecture for vehicle companies and is a validated platform, focuses on high reliability and safety for automotive systems. However, ROS2, which is widely used in the field of robotics, lacks safety and real-time control but offers substantial advantages in terms of flexible communication, development convenience through an active open-source community, the development of sensor drivers, and more. In addition, ROS2 has notable strengths in powerful visualization, development tools, and simulation tools. The integration of Adaptive AUTOSAR and ROS2 leverages the safety of the automotive field and the flexibility of robotics to simplify and accelerate the development and testing of autonomous vehicles. In addition, interoperability via the SOME/IP protocol enables faster adoption in the current automotive industry compared to DDS. To integrate these two distinct platforms, this study proposes the design and implementation of an interoperable architecture named Autonomous Driving System with Integrated ROS2 and Adaptive AUTOSAR (ASIRA). Using ASIRA within a Linux environment enables the exchange of data between Autoware, which is ROS2's autonomous driving project, and the Adaptive AUTOSAR Platform, which facilitates the operation of autonomous vehicles.

The structure of this paper is as follows: Section 2 introduces related research and background knowledge. Section 3 describes the system architecture and components and the method of system implementation. Section 4 validates the developed system through the simulation within scenarios and verifies the capability of the two platforms to exchange data and achieve autonomous driving. Finally, Section 5 concludes this paper and presents future research directions.

## 2. Background and Related Works

Section 2 provides background information on the platforms and technologies used in this thesis and other related works. Section 2.1 describes Adaptive AUTOSAR. Section 2.2 describes ROS and ROS2. Section 2.3 describes related works and discusses their advantages and limitations.

### 2.1. Adaptive AUTOSAR

Released in 2006, AUTOSAR 2.0 has since evolved to AUTOSAR 4.3 (2017) and is widely used in a variety of vehicles. However, the recent rise of electric vehicles and autonomous driving has placed new demands on the existing AUTOSAR platform, such as high network bandwidth and high computing power. To meet these demands, researchers have attempted to install Linux in vehicles. A Linux environment compensates for some of the disadvantages of the existing AUTOSAR platform but lacks essential software that is used in vehicle development environments, such as CAN communication and diagnostic functions. Against this backdrop, the AUTOSAR consortium has labeled the original AUTOSAR platform as the Classic Platform and has introduced the POSIX OS-based Adaptive AUTOSAR Platform.

The overall architectural logical view of Adaptive AUTOSAR is shown on the left side of Figure 1 [18]. Adaptive Applications (AAs) are located in the top layer. AAs run on AUTOSAR Runtime for Adaptive Applications (ARAs). ARAs consist of Platform Foundation Functional Clusters (FCs), and they also include Adaptive Platform Services and Standard Application/Interfaces, although they are not shown in the figure. FCs

provide the fundamental functions of Adaptive AUTOSAR. Additionally, as the AP release continues to evolve, new FCs are being added.

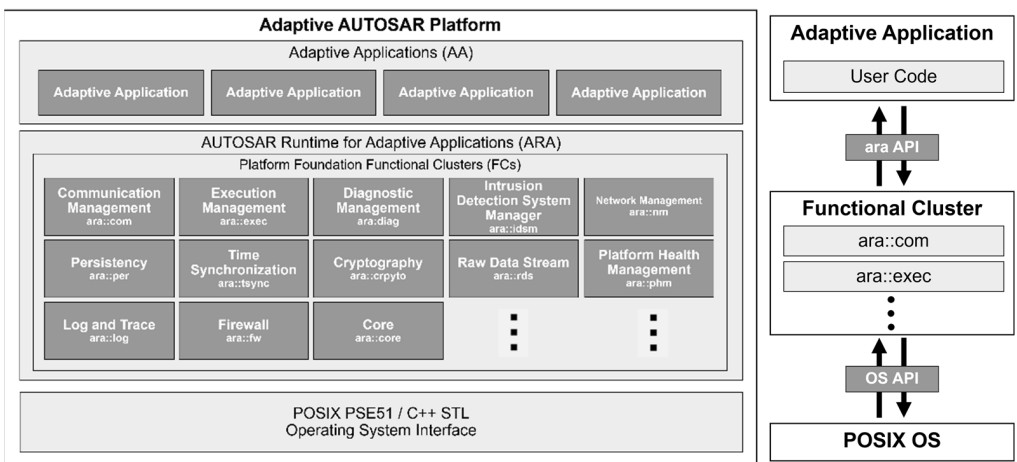

**Figure 1.** Architecture (**left**) and operating mechanism (**right**) of Adaptive AUTOSAR Platform.

The operating mechanism of the Adaptive AUTOSAR Platform is depicted on the right side of Figure 1. It includes an FC that uses the POSIX-based OS API, within which the AUTOSAR Runtime exists. Using this API, users can develop and use the necessary applications. Similarly, in this study, applications using ARAs are developed to implement integration with ROS2. A considerable difference between Adaptive AUTOSAR and the traditional Classic Platform is that it does not rely on traditional signal-oriented communication but is based on SOC. By employing SOME/IP implemented within ara::com based on SOC, applications can be developed that allow for dynamic connections between servers and clients.

### 2.2. ROS

#### 2.2.1. ROS 1

Since the publication of a paper on ROS open-source middleware in 2009 [19], ROS middleware has been widely used in robotics and autonomous system development by universities, research organizations, and individuals. In particular, the community is quite active, which has brought about the release of numerous libraries and packages for various types of robots as open-source (for example, Point Cloud Library (PCL) [20]). Moreover, drivers for sensors that are produced by various companies are also available, and this has contributed to the rapid growth of the ROS ecosystem. However, ROS1, which was developed for the purpose of research, encountered various limitations in commercial use. For instance, the development of NASA's Robonaut was based on ROS1, but because of the need for real-time control, ROS1 was modified during development. In addition to real-time control, there were several shortcomings from an industrial perspective, including OS limitations, single points of failure, and security issues.

#### 2.2.2. ROS2

In 2017, Open Robotics distributed the first version of ROS2 to supplement the deficiencies of ROS1 and to enable its use in commercial applications. One pivotal change that appeared in ROS2 was the decision not to use TCPROS, which was the norm in ROS1, but instead to adopt DDS as the fundamental middleware for internode communication. This service is designated as an international standard by the Object Management Group [21], and it allows the integration of a security specification known as DDS-Security into ROS2. This enhancement addressed the security vulnerabilities in ROS1. Additionally, the capability of DDS to use RTPS meant that it could also fulfill real-time control requirements.

In ROS2, a Robot Client Library (RCL) is available. It supports various programming languages under names such as rclcpp, rclpy, and rcljava. This substantially enhances development flexibility relative to ROS1 and allows for selecting an optimized language for specific algorithms.

### 2.3. Related Works

Recent studies have been conducted on the interoperability between Adaptive AUTOSAR and ROS2, and on the application of ROS2 in vehicles. Jacqueline Henle compared ROS2 and Adaptive AUTOSAR to evaluate their suitability for future vehicle architectures [22]. Adaptive AUTOSAR, which applies Service-Oriented Architecture and Ethernet, provides wider bandwidth, and this makes it suitable for the architecture of future vehicles such as autonomous cars. However, it also has several disadvantages. In terms of ara::iam security, platform health checks, persistence of information through ara::per, and vehicle diagnostic functions, Adaptive AUTOSAR is superior to ROS2. However, in terms of maintaining a continuous integration and development environment, loose coupling between nodes, development speed, and the provision of powerful debugging and visualization tools, ROS2 was evaluated as being better. However, the comparison did not consider the interoperability between the two platforms. This study proposes the design of an architecture that compensates for the disadvantages of both platforms while enhancing their advantages to enable interaction between them, and this is validated through autonomous driving simulation.

Arestova presented a new architecture that met the service-oriented and real-time communication needs of the automotive industry by integrating AUTOSAR Adaptive, Open Platform Communications Unified Architecture (OPC UA), and Time-Sensitive Networking (TSN) technologies [23]. Adaptive AUTOSAR supports dynamic deployment and high-performance processing, OPC UA enables flexible SOC between devices, and TSN ensures real-time network communication. This research investigated how a combination of these three technologies could provide high-speed deterministic communication, particularly regarding how OPC UA's Client-Server and PubSub models could be applied to the AUTOSAR Adaptive SOC. In addition, the effectiveness of this architecture was verified through the implementation of bindings for real-time systems and performance evaluation. However, the researchers performed only the implementation and verification of the communication itself and did not conduct testing in actual operational environments. In this study, an autonomous driving scenario was used to verify the interoperability of the two platforms, and the developed system was validated through simulation.

## 3. System Architecture and Components

Section 3 describes the structure and methods of implementation for the ASIRA architecture developed in this study. Section 3.1 explains the overall architecture, the roles of each component, and the flow of data. Section 3.2 discusses the methods used to implement the architecture.

### 3.1. System Architecture

The system architecture of this study is divided into two main parts: the Adaptive AUTOSAR Platform and the ROS2 Autonomous Driving Platform, as shown in Figure 2.

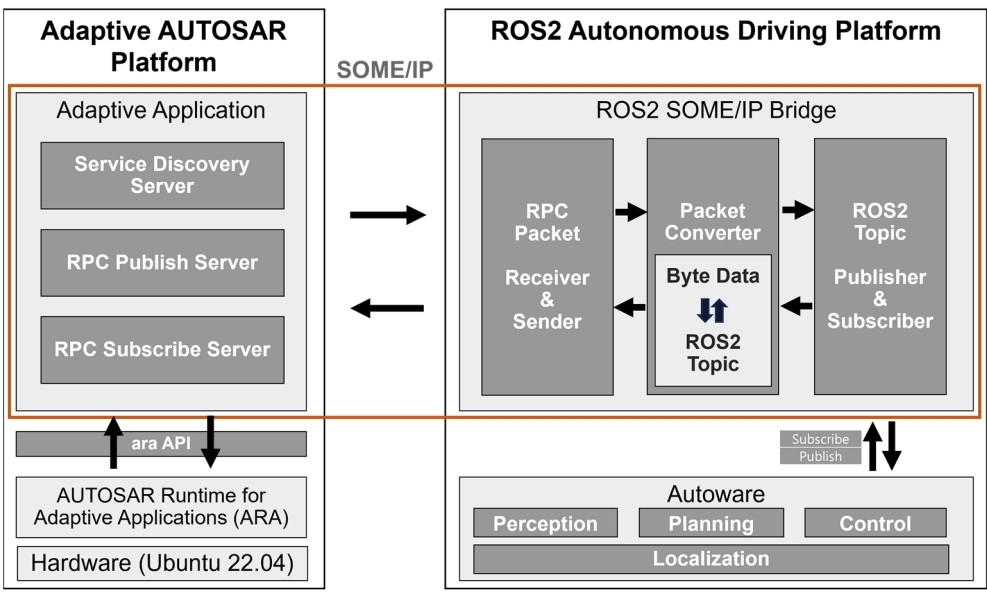

**Figure 2.** ASIRA system's architecture.

### 3.1.1. Adaptive AUTOSAR Platform

The Adaptive AUTOSAR Platform used open-source software developed in compliance with the AUTOSAR standard to simulate the Adaptive AUTOSAR Runtime on Linux [24]. It was implemented based on R20-11 and R22-11. Not all the components of ARA were implemented; however, as explained in the figure, ara::com, ara::core, ara::diag, ara::exec, ara::log, and ara::phm were implemented in accordance with the AUTOSAR standard. AAs can develop the functionalities they need using ARAs, which complies with the AUTOSAR standard. In this study, we developed an AA based on ARAs that could exchange the necessary data packets with the ROS2 Autonomous Driving Platform by constructing a SOME/IP Service Discovery (SD) Server and an RPC Publish/Subscribe Server.

### 3.1.2. ROS2 Autonomous Driving Platform

The ROS2 Autonomous Driving Platform section was implemented using Autoware, which is an open-source autonomous driving software platform based on ROS2, and an ROS2 SOME/IP Bridge proposed in this study. Autoware includes all the necessary components for operating autonomous vehicles, such as sensing, localization, perception, planning, and control. It also provides its own simulation environment. In this study, the Autoware's simulation environment was used to implement and verify the integration between the Adaptive AUTOSAR Platform and ROS2.

The ROS2 SOME/IP Bridge consists of three main parts. The first part contains the Receiver and Sender, which are responsible for receiving and transmitting RPC Request/Response packets for integration with the Adaptive AUTOSAR Platform. The second part involves converting the received packets into "Data Types" that can be used in ROS2, transforming them into the required Topic Name and Message Type, and generating messages, or, conversely, converting them into byte arrays that can be packed into RPC Request/Response packets. The last part publishes the converted messages to the ROS2 Platform via DDS middleware or receiving topics that need to be transmitted from Autoware to the Adaptive AUTOSAR Platform. The nodes existing within the ROS2 SOME/IP Bridge are all structured to fulfill the functions of these three parts, and they can be modified as required according to the format or type of data.

### 3.1.3. Data Flow

Figure 3 presents the data flow diagram for the entire system. This represents the implementation within the ROS2 SOME/IP Bridge to exchange Kinematic State and control

commands as part of the actual implementation and verification of the ASIRA system's architecture. The execution sequence is indicated by the numbers.

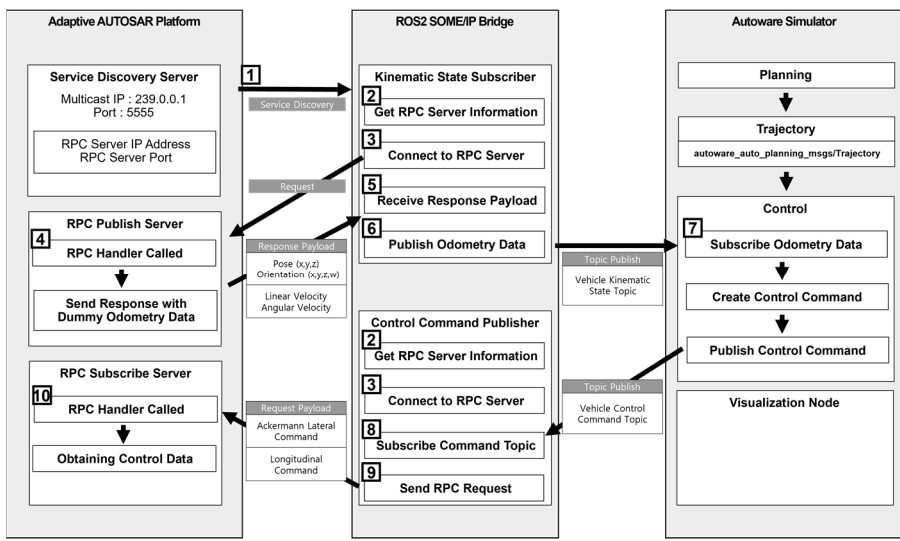

**Figure 3.** Data flow.

1. The SD Server opens a server with the multicast IP and Port parsed from ARXML, and the endpoint information of the RPC Servers. The Kinematic State Subscriber and Control Command Publisher nodes within the ROS2 SOME/IP Bridge join the multicast group and send a findService packet. The SD Server checks the Service ID information in the packet and sends the corresponding RPC endpoint information to the Bridge Nodes.

2. From the RPC Endpoint packet received from the SD Server, each packet obtains the respective RPC server endpoint information. The Kinematic State Subscriber acquires the Endpoint of the RPC Publish Server, whereas the Control Command Publisher obtains the Endpoint of the RPC Subscriber Server.

3. Bridge Nodes attempt to connect to the RPC Server using the acquired endpoint information. After the connection is established, they send a request.

4. When an RPC Request is received from the Kinematic State Subscriber, the Handler of the RPC Publish Server is called. The Handler generates Dummy Vehicle Odometry Data for system verification and sends it back to the Kinematic State Subscriber in the Response Payload.

5. Upon receiving the Response Payload, the Kinematic State Subscriber analyzes the packet to obtain information on Pose, Orientation, Linear Velocity, and Angular Velocity.

6. This information is then formatted according to the nav_msgs/Odometry Message used in ROS2 and published on the ROS2 Platform. This includes the vehicle's position, direction, and speed.

7. The ROS2 Autoware platform subscribes to the Odometry Topic. Subsequently, it acquires Trajectory information from the Planning Node and sends it to the Control Node. The Control Node uses this information to generate a Vehicle Control Command and publishes the Topic.

8. The Control Command Publisher Node subscribes to the Vehicle Control Command Topic and converts it into a byte array according to the RPC Request Payload format to create a packet.

9. It sends a request to the RPC Subscribe Server.

10. When the RPC Subscribe Server receives the request, it extracts the Vehicle Control Command from the Payload, which includes Lateral and Longitudinal Commands with details such as steering angle, steering rotation rate, speed, acceleration, and jerk values. This information is then converted into a format that is usable by the Adaptive AUTOSAR Platform.

This is how the entire system flow unfolds. The data sent from the Adaptive AUTOSAR Platform are used by the ROS2 Autonomous Driving Platform via the Bridge Node. The resulting values from the processing of those data are then passed back to the Adaptive AUTOSAR Platform via the Bridge for use.

*3.2. System Implementation*

The system implementation consists of the ARXML creation of the Adaptive AUTOSAR Platform, the Adaptive Application implementation, and the ROS2 SOME/IP implementation.

3.2.1. Writing ARXML for the Adaptive AUTOSAR Platform

The typical development process for the Adaptive AUTOSAR Platform is to create an ARXML Manifest file and design an AUTOSAR modeling and E/E architecture based on the Manifest file. Typically, this is completed with a toolchain from companies that offer different AUTOSAR tools. In this study, we do not use an AUTOSAR solution from any specific company, and we use ARXML to define the various arguments needed to run the application. The specification for the application to be executed is written and passed as an argument through the ARXML Manifest file, and the argument is then used to execute the Adaptive AUTOSAR Platform. First, Execution Management via ara::exec is executed, and then the AA, which is responsible for interfacing with Execution Management ROS2, is executed.

The AA reads an ARXML Manifest file through the ARXML Reader. The file contains the parameters for setting up the SOME/IP network. The Manifest file used in this study follows the standard R20-11 Specification of Manifest for Adaptive Platform [25], and its main properties are shown in Table 1.

**Table 1.** Communication cluster arguments.

| Component Name | Description |
| --- | --- |
| COMMUNICATION CLUSTER | A communication cluster is the main element that describes the topological connection of ECUs connected by a communication medium. Nodes within a cluster share the same communication protocol. A communication cluster has one or more physical channels. |
| ETHERNET-PHYSICAL-CHANNEL | Ethernet/physical channels represent VLANs or untagged channels. |
| NETWORK-ENDPOINTS | A network endpoint defines an IP address or MAC multicast address, for example. |
| PROVIDED-SOMEIP-SERVICE-INSTANCE | The existence and configuration of service instances implemented on top of SOME/IP. |
| PROVIDED-EVENT-GROUPS | For each event group, configure the communication settings for the service instance. |
| SD-SERVER-CONFIG | Configure settings related to the SD Server. |
| INITIAL-OFFER-BEHAVIOR | Configure settings related to the server's OFFER BEHAVIOR and the client's FIND BEHAVIOR. |
| INITIAL-DELAY-MIN-VALUE | Sets the minimum time (in seconds) to randomly delay the first OFFER BEHAVIOR. This applies to the SD Server's Initial Offer or the SD Client's find message. |
| INITIAL-DELAY-MAX-VALUE | Sets the maximum time (in seconds) to randomly delay the first OFFER BEHAVIOR. This applies to the SD Server's Initial Offer or the SD Client's find message. |

Table 1 describes the main components of implementation of an Adaptive Application (AA). The AA that is used to integrate ROS2 has three main components.

The first is the communication cluster component. In the cluster configuration, we set up two Endpoints (EPs), which are ServiceDiscoveryEP and RPCSubscribeServerEP. ServiceDiscoveryEP is the endpoint information for SD, and RPCSubscribeServerEP is the endpoint information for RPC communication after a connection is configured through SD.

The second is the Ethernet Communication Connector. It defines the properties of the EPs, which are the components of the communication cluster mentioned above and is used in this study by specifying only the port number.

Finally, we configure the Provided SOME/IP Service Instance. This is used by the AA to configure and set up the SOME/IP SD Server for mutual discovery with the ROS2 Bridge. One SD Server is running, and we specify the IP and Port in the appropriate band to enable Multicast IP. We then apply a maximum/minimum latency time for the SD Server's Initial Offer.

### 3.2.2. Implementation of Adaptive Applications on an Adaptive AUTOSAR Platform

Execution management starts the AA on a new thread. On the first run, the AA imports the ARXML Manifest file and parses the parameters to set up the SOME/IP SD Server and RPC server. It creates the SD Server through the SOME/IP inside ara::com and waits while it is connected with the network group having the multicast group IP and address. At this time, it enters the waiting state before sending the first message based on the Initial Delay, as configured above. Then, the Consumer (ROS2 SOME/IP Bridge in this case) configures a packet for the desired service and delivers it to the multicast group through the findService message. At this time, the SD Server checks the Service ID information in the packet, and if it matches the Service ID of the server, it sends a service offer message with endpoint information to the multicast group. This process is called SD and allows the Adaptive AUTOSAR Platform and ROS2 SOME/IP Bridge to find each other and to obtain endpoint information from the consumer side to request a connection.

RPC Servers are created from the endpoint information parsed from the ARXML Manifest file. We used the RPCs supported by the ara::com internal SOME/IP network binding. We created a total of two RPC servers. The roles of the created RPC servers are as follows.

RPC Publish Server: When a request from the ROS2 Bridge comes in, the Adaptive AUTOSAR Platform sends the Vehicle's Odometry Data to the ROS2 Bridge in an RPC Response packet.

RPC Subscribe Server: When a request comes from the ROS2 Bridge, the RPC Subscribe Server analyzes the request packet to obtain the command information for the vehicle and makes the information available to the Adaptive AUTOSAR Platform.

We have two RPC servers and separate connections to the ROS2 Bridge because each data transmission cycle is different. If we have two RPC servers, each server can send and receive data independently of the other. If there is a delay in sending a request to the RPC server, the response will also be delayed, which can be fatal for a system that sends and receives real-time data.

After we created the RPC server, we created an RPC Handler for each server and bound each Handler to the RPC Server. The binding process involves registering the Service ID and Method ID with the RPC server. Each Handler is responsible for checking received packets, analyzing the payload of the packet, and either consuming it or sending an appropriate response. The Handlers connected to the two RPC servers play the following roles.

RPCKinematicStateHandler: This Handler is associated with the RPC Publish Server. When an RPC request comes from the ROS2 SOME/IP Bridge, the RPCKinematicStateHandler is called. This is the Handler that sends the Vehicle's Odometry Information to the ROS2 Autoware. It generates packets that contain Dummy Vehicle Odometry Information generated during the simulation and sends them to the RPC Response. The data to be transmitted are shown in Table 2.

**Table 2.** Data Component.

| Component Name | Description |
|---|---|
| position_x, y, z | Contains the current location of the vehicle. Sent in the UTM coordinate system used by the Autoware Simulator. |
| orientation_x, y, z, w | Information indicating the direction of the vehicle. Represented as a quaternion. |
| linearVelocity_x, y, z | Information about the vehicle's Linear Velocity along the X, Y, and Z axes. |
| angularVelocity_x, y, z | Information about the vehicle's Angular Velocity along the X, Y, and Z axes. |

RPCVehicleCommandHandler: This Handler is associated with the RPC Subscribe Server. When an RPC request comes from the ROS2 SOME/IP Bridge, RPCVehicleCommandHandler is called. This Handler processes the Vehicle Odometry Information that is sent by the RPCKinematicStateHandler in the ROS2 Autoware, and it receives a packet containing a command to follow the vehicle's trajectory as a request. It extracts the information from the request packet to be used by the Adaptive AUTOSAR Platform. The data received are shown in Table 3.

**Table 3.** Vehicle Control Command Data Component.

| Component Name | Description |
|---|---|
| steering_tire_angle | Represents the steering angle, which is the angle between the front wheels of the vehicle and the vehicle body. |
| steering_tire_rotation_rate | Rotational speed of the steering wheel. |
| acceleration | Acceleration of the vehicle. |
| jerk | Rate of change in the vehicle's acceleration. |

3.2.3. Implementation of ROS2 SOME/IP Bridge

After it receives the packet, ROS2 SOME/IP Bridge extracts the endpoint information from the packet for the server to which the user wants to connect to. The endpoint information contains the IP address and port information for the server, which constitutes a client socket for SOME/IP RPC. The ROS2 SOME/IP Bridge contains two types of sockets.

Figure 4 shows the components of a multicast data packet that the ROS2 SOME/IP Bridge receives to locate an SD Server and follows the R20-11 release AUTOSAR standard [26]. Among the components of this packet, the data that are primarily used by the ROS SOME/IP Bridge are shown in Table 4.

**Table 4.** SOME/IP Service Discovery Packet components.

| Type | SD Type |
|---|---|
| Service ID | Service Unique ID in vehicle |
| Instance ID | Instance ID under service |
| IPv4/6 Address | Endpoint IP Address of SOME/IP Server |
| L4-Proto | Transport Layer Protocol |
| Port Number | Endpoint Port of SOME/IP Server |

**SOME/IP Service Discovery Packet**

| 0 | 1 | 2 | 3 | 4 | 5 | 6 | 7 | 8 | 9 | 10 | 11 | 12 | 13 | 14 | 15 | 16 | 17 | 18 | 19 | 20 | 21 | 22 | 23 | 24 | 25 | 26 | 27 | 28 | 29 | 30 | 31 |
|---|---|---|---|---|---|---|---|---|---|---|---|---|---|---|---|---|---|---|---|---|---|---|---|---|---|---|---|---|---|---|---|

| Flags [8 bit] | | | Reserved [24 bit] | | | |
|---|---|---|---|---|---|---|
| Length of Entries Array in Bytes [32 bit] | | | | | | |
| Type | | Index 1st options | Index 2nd optins | | # of opt1 | # of opt2 |
| Service ID | | | Instance ID | | | |
| Major Version | | TTL | | | | |
| Minor Version | | | | | | |
| Type | | Index 1st options | Index 2nd optins | | # of opt1 | # of opt2 |
| Service ID | | | Instance ID | | | |
| Major Version | | TTL | | | | |
| Minor Version | | | | | | |
| Length of Entries Array in Bytes [32 bit] | | | | | | |
| Length | | | Type | | Reserved | |
| IPv4-Address | | | | | | |
| Reserved | | L4-Proto | | Port Number | | |

**Figure 4.** SOME/IP Service Discovery Packet.

After receiving the packet, the ROS2 SOME/IP Bridge extracts the endpoint information from the packet for the server to which the user wants to connect to. The endpoint information contains the IP address and port information for the server, which constitutes a client socket for SOME/IP RPC. The ROS2 SOME/IP Bridge contains two types of sockets.

KinematicStateSubscriber: Continuously sends an RPC Request to the RPC Publish Server of the Adaptive AUTOSAR Platform to request the Kinematic State. The Adaptive AUTOSAR Platform receives the request and sends information indicating the current odometry of the vehicle in the response. The Bridge analyzes the Response packet, extracts the required information, and enters the data according to the nav_msgs/Odometry Message type of the ROS2. The nav_msgs/Odometry message is as shown in Table 5.

**Table 5.** Odometry Message components.

| Topic Type | Description |
|---|---|
| geometry_msgs/PoseWithCovariance pose | Represents the current position and Orientation (Pose) of the vehicle. A Pose consists of a position (X, Y, Z) and a direction (quaternion). |
| geometry_msgs/TwistWithCovariance twist | Fields representing the Linear and Angular Velocities of the vehicle, including the speed of travel and rotation. Linear and Angular Velocities are organized as three-dimensional vectors. |

We populate the Odometry Message with the data extracted from the packet and publish it on the ROS2 Platform to be used by Autoware.

ControlCommandPublisher: Subscribe to the Vehicle Control Command Topic published by Autoware based on the ROS2 Platform. It has a steering wheel and a speed profile that allows the vehicle to follow the trajectory generated by the Planning algorithm in the autonomous driving platform. Data are extracted from this Topic and sent in an RPC Request packet to the RPC Subscribe Server of the Adaptive AUTOSAR Platform. The Vehicle Control Command Topic is structured as shown in Table 6.

**Table 6.** Vehicle Control Command Topic components.

| Topic Name | Description |
|---|---|
| AckermannLateralCommand | Represents the vehicle's lateral control commands based on the Ackermann drive mechanism. This includes commands related to the steering angle of the vehicle. |
| LongitudinalCommand | Represents commands for the longitudinal control of the vehicle. This includes commands related to vehicle speed, acceleration, and braking. |

Based on the data extracted from the Topic, ControlCommandPublisher constructs a packet to be sent to the Adaptive AUTOSAR Platform. The data content of the AckermannControlCommand Topic is then sent as a payload in an RPC Request to make the vehicle control information available to the Adaptive AUTOSAR Platform.

## 4. System Validation

Section 4 describes the procedure for validating the ASIRA architecture built in this paper. Section 4.1 describes the configuration of the environment to verify the system. Section 4.2 describes the verification scenario and simulation method. In Section 4.3, we show that the ASIRA architecture built in this study can be interoperable based on the actual simulation results.

### 4.1. System Verification Environment

To verify the system that was implemented in this study, we created a Point Cloud Map and a Vector Map for use on the ROS2 Autonomous Driving Platform. The red square in Figure 5 shows the area where the LiDAR sensor was used to collect data as the robotic platform moved to collect the data shown in Figure 6. While it collected 16-channel 3D LiDAR data, the robot constructed a Point Cloud Map by performing gicp/ndt scan matching and Simultaneous Localization and Mapping (SLAM) using a graph-based SLAM algorithm [27].

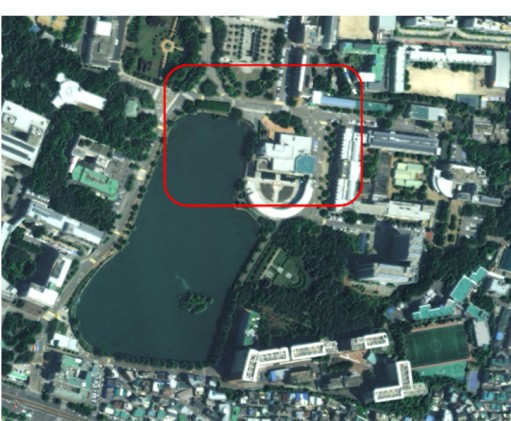

**Figure 5.** Scenario place.

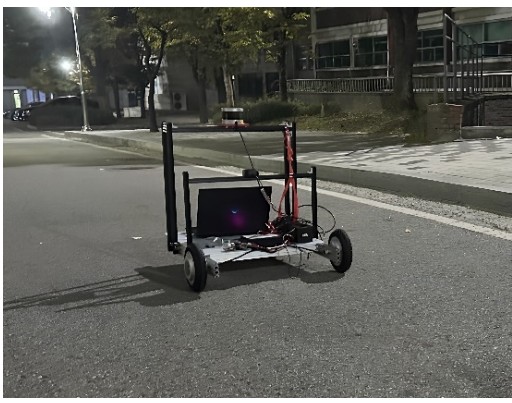

**Figure 6.** Sensor data acquisition robot.

The results of the SLAM are shown in Figure 7. We created a three-dimensional map of points, which was used in the Localization and Perception process in Autoware. Along with the Point Cloud Map, a Vector Map was constructed, and it is shown in the figure. The Vector Map used a Lanelet2 format that contained the location information for the road on which the vehicle could drive, such as left and right lanes, stop lines, traffic lights, and the various constraints required for driving. Using TIER V4's Vector Map Builder [28], we constructed a Vector Map containing road information for use in the verification of the simulation. In Figure 8, the yellow-colored part of the overlapping Point Cloud is the area where the vehicle could drive.

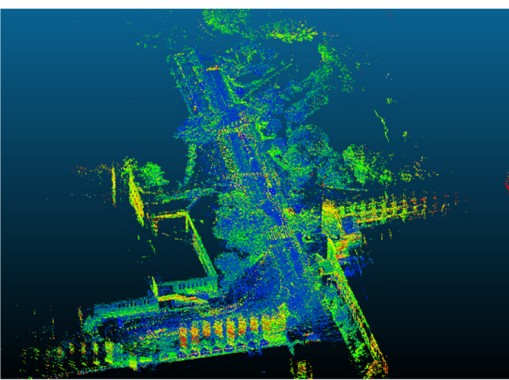

**Figure 7.** Three-dimensional Point Cloud Map.

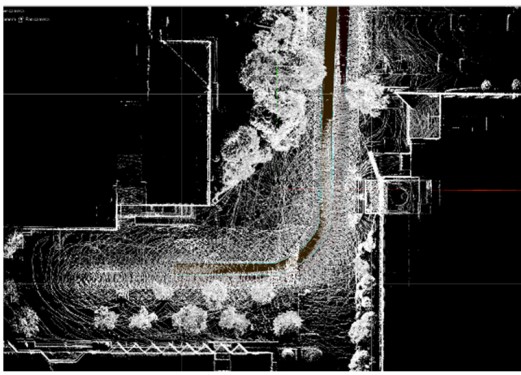

**Figure 8.** Point Cloud Map with Vector Map.

*4.2. Validation Scenarios*

Figure 9 shows a scenario that was used to validate the system proposed in this study. Below, we describe the implementation method for the verification of each part.

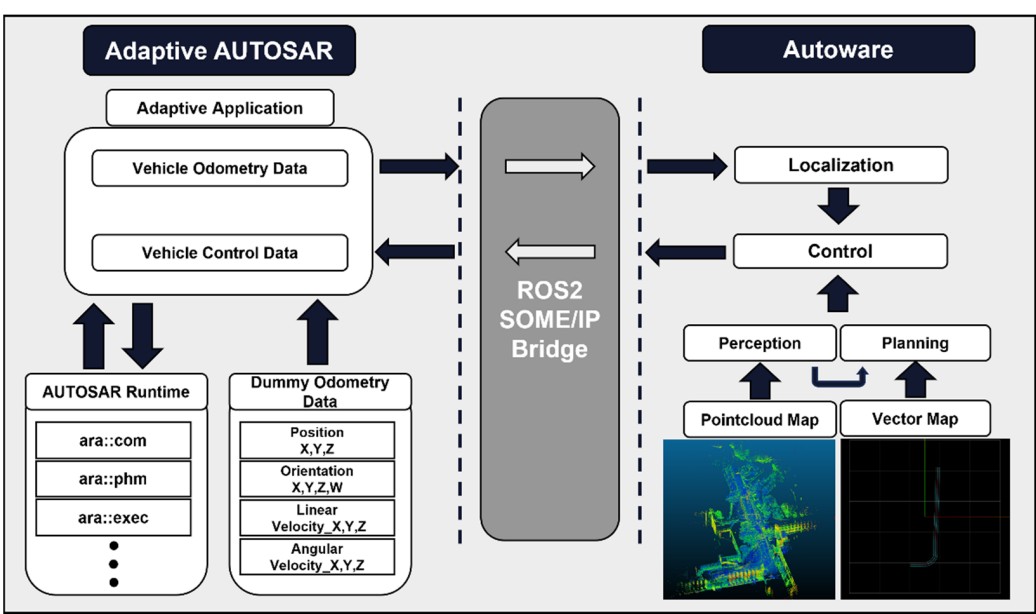

**Figure 9.** System validation scenario.

Dummy Odometry Data File: The validation scenario assumes that a vehicle knows its Position, Orientation, and Linear and Angular Velocities. The Adaptive AUTOSAR Platform is equipped with Dummy Odometry Data. Dummy location information was obtained by operating the actual data acquisition robot shown in Figure 6 in the scenario environment. The odometry results obtained during this process were recorded at a 20 ms interval along with Timestamps. This information was then converted into a CSV file for use in the verification process of ASIRA.

Figure 10 shows some of the acquired Odometry Data converted to a CSV file with a timestamp. Some Linear Velocities and accelerations are omitted for readability in Figure 10. When a request is received from ROS2 SOME/IP Bridge, the CSV file is read row by row to simulate the acquisition of the vehicle's location information. The X, Y, and Z positions of the vehicle are based on the UTM coordinate system using a latitude of 37.5422 and a longitude of 127.0785 as the origin. Odometry Data also contain Orientation X, Y, Z, and W values to indicate the Orientation of the vehicle, as well as Linear Velocity and Angular Velocity.

| Timestamp | Frame ID | Child Frame ID | Position X | Position Y | Position Z | Orientation X | Orientation Y | Orientation Z | Orientation W | Linear Velocity X |
|---|---|---|---|---|---|---|---|---|---|---|
| 1706593812 | map | base_link | 27.31285 | −0.50689 | −0.54082 | 0.0000000 | −0.00377 | 0.0000000147095 | 0.999993 | 0.000434 |
| 1706593812 | map | base_link | 27.31288 | −0.50689 | −0.54082 | 0.0000000 | −0.00377 | 0.0000001299158 | 0.999993 | 0.002161 |
| 1706593812 | map | base_link | 27.31297 | −0.50689 | −0.54082 | 0.0000000 | −0.00377 | 0.0000005088407 | 0.999993 | 0.005558 |
| 1706593812 | map | base_link | 27.31317 | −0.50689 | −0.54082 | 0.0000000 | −0.00377 | 0.0000012592063 | 0.999993 | 0.010162 |
| 1706593812 | map | base_link | 27.31349 | −0.50689 | −0.54082 | 0.0000000 | −0.00377 | 0.0000024870027 | 0.999993 | 0.015873 |
| 1706593812 | map | base_link | 27.31397 | −0.50689 | −0.54082 | 0.0000000 | −0.00377 | 0.0000042362871 | 0.999993 | 0.022391 |
| 1706593812 | map | base_link | 27.31462 | −0.50689 | −0.54082 | 0.0000000 | −0.00377 | 0.0000065436845 | 0.999993 | 0.029553 |

**Figure 10.** Dummy Odometry Data file.

ROS2 SOME/IP Bridge: As described in Section 3.2, the ROS2 SOME/IP Bridge is responsible for requesting and receiving vehicle location data and sending it to the ROS2 Topic. It receives the Vehicle Command Topic from Autoware and sends it to the Adaptive AUTOSAR Platform via an RPC request. Also, it requests location information from the Adaptive AUTOSAR Platform every 2.5 ms to match the rate of the odometry information used by Autoware.

Autoware: When Autoware receives odometry information from the ROS2 SOME/IP Bridge, it uses the information obtained from localization. It also performs Perception using the Point Cloud Map, performs Planning using Perception and Vector Map results, and generates Trajectory information. This is used by the Control Node to finally publish the Vehicle command Control Topic. In addition to the algorithmic part, the simulation environment is configured through visualization using ROS2 Rviz.

Figure 11 shows the application of the Point Cloud Map and Vector Map for system validation using Rviz, a simulator and visualization tool provided by Autoware. In the Vector Map, the green Lanelet2 component represents the route that the vehicle can drive. The shortest distance to the destination is calculated and the estimated route is displayed in green. In the figure, the estimated route is currently displayed as a green line. The Global Path is displayed in light green, and the possible route based on the current vehicle position is displayed in dark green. You can set the initial position of the vehicle using the 2D Pose Estimation button and set the destination of the vehicle using the 2D Goal Pose button.

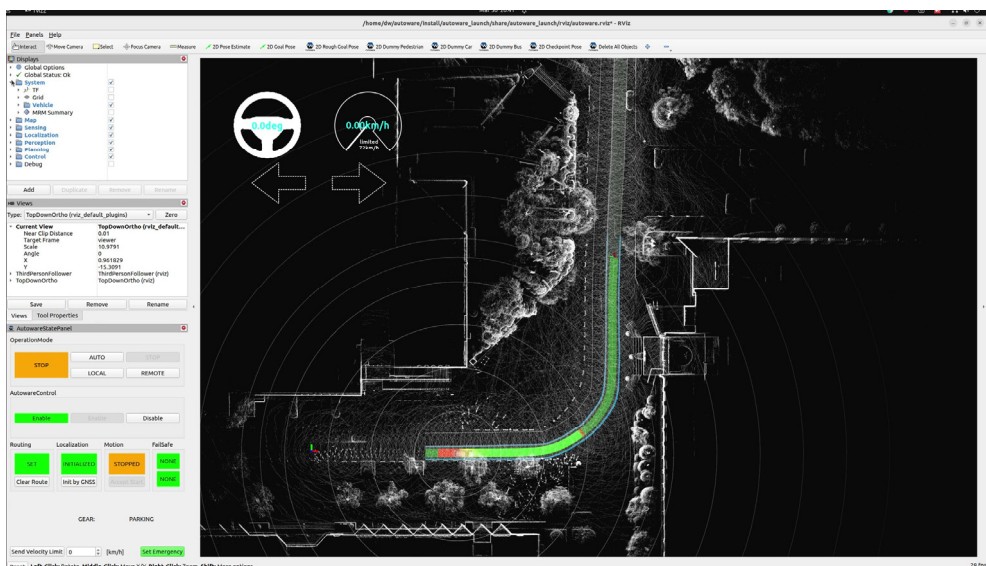

**Figure 11.** Autoware Simulator with 3D Point Cloud and Vector Map.

*4.3. System Verification*

Figure 12 shows a simulation of the entire system. The top left terminal is the Kinematic State Subscriber and the top right terminal is the Vehicle Command Publisher. The bottom terminal is the Adaptive AUTOSAR Platform. On the right side of the figure is the Rviz screen for visualizing the Autoware simulation.

Figure 13 shows the change in the trajectory and path over time. The Adaptive AUTOSAR Platform receives the Dummy Odometry Information transmitted by ROS2 over time and continuously drives to the destination, and the path of the vehicle is continuously calculated and updated accordingly. This means that ROS2 and the Adaptive AUTOSAR Platform are working together and exchanging data in real time. (Odometry Data should be delivered within 20 ms to meet Autoware's needs. Vehicle Control Commands should be delivered within 50 ms to match Autoware's cycle.)

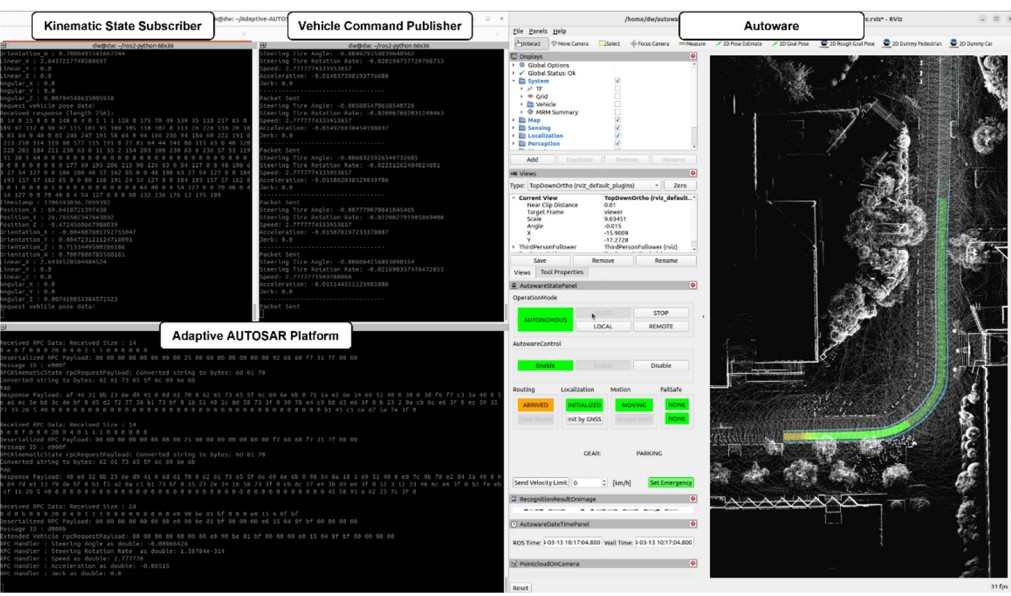

**Figure 12.** Full system simulation.

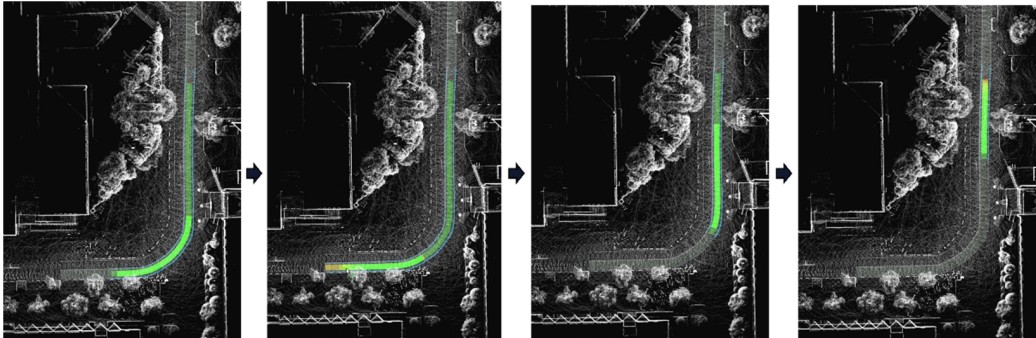

**Figure 13.** Changes in Trajectory and Path over Time.

Figure 14 displays a terminal outputting data exchanged during a simulation. In the top left of the figure is the Kinematic State Subscriber, which shows data exchange on Terminal 1. It receives the vehicle's Position, Orientation, Linear Velocity, and Angular Velocity from the Adaptive AUTOSAR Platform and converts them into ROS2 messages for publishing. As shown on Terminal 1, it receives packets containing the vehicle's position X, Y, and Z, Orientation X, Y, Z, and W, along with Linear and Angular Velocity, and analyzes the payload to produce results.

The top right of the figure is the Vehicle Command Publisher, which receives the Vehicle Control Command topic from Autoware and sends it to the Adaptive AUTOSAR Platform. As displayed on Terminal 2, it captures Steering Tire Angle, Rotation Rate, Speed, Acceleration, and Jerk information included in the Control Command topic before sending the packet.

The bottom of the figure represents the Adaptive AUTOSAR Platform, which interfaces with Autoware's autonomous driving software through ROS2 Bridge for data transmission. Terminal 3 shows the platform sending Odometry Data packaged in RPC Response Payload and receiving Vehicle Command packets through Request Payload. The RPC Handler parses these packets to display Steering Angle, Rotation Rate, Speed, Acceleration, and Jerk information in double format. It also illustrates the process of receiving current location information from Odometry Dummy Data, serializing it, and transmitting it in the Response Payload.

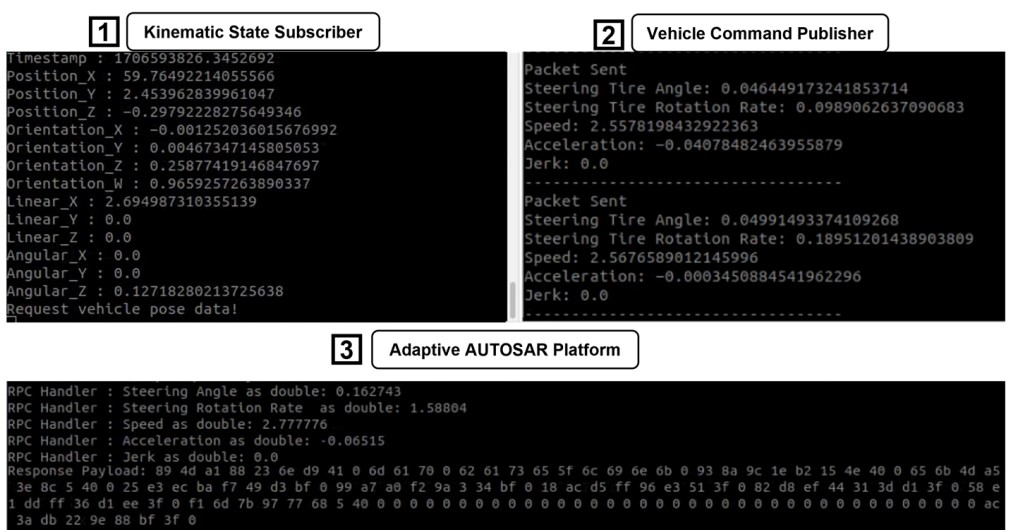

**Figure 14.** Terminal screen of Adaptive AUTOSAR Platform and ROS2 SOME/IP Bridge.

Table 7 documents the details of two types of data transmitted in a scenario, measured over a 3 min driving session. The delay was measured from the moment data were generated—through serialization, transmission via the ROS2-SOME/IP Bridge—to the conversion of the packet into a usable data format. The Odometry Data transmitted from the Adaptive AUTOSAR Platform to the Kinematic State Subscriber was aligned with the 50 Hz (20 ms) transmission cycle required by the Autoware autonomous driving platform. The average and peak delay times were recorded at 10.95 ms and 13.45 ms, respectively. These levels are significantly below the 20 ms cycle, indicating that they are sufficiently low for use in the autonomous driving platform proposed in this study. Data transmission from the Vehicle Command Publisher to the Adaptive AUTOSAR Platform also demonstrated very low levels compared to the cycle, proving its suitability for use in the autonomous driving platform described in this research.

**Table 7.** Analysis of average and peak delay times for data transmission in system simulation.

| From | To | Frequency | Average Delay | Peak Delay |
| --- | --- | --- | --- | --- |
| Adaptive AUTOSAR Platform | Kinematic State Subscriber | 50 Hz | 10.95 ms | 13.45 ms |
| Vehicle Command Publisher | Adaptive AUTOSAR Platform | 20 Hz | 5.19 ms | 8.77 ms |

## 5. Conclusions and Future Improvements

In this study, we built an architecture that allowed Adaptive AUTOSAR and ROS2 to be interconnected through the Ethernet-based SOME/IP protocol and presented the simulation results. We showed that it was possible to connect ROS2-based autonomous driving and Adaptive AUTOSAR-based vehicle architecture, which have been studied in different fields. The vehicle location information from the Adaptive AUTOSAR Platform is transmitted to the ROS2 Bridge Node using the SOME/IP protocol, which is converted to an ROS2 Topic that can be subscribed to by the autonomous driving system. In addition, instead of using only one-way data transmission, the ROS2 Platform uses received data to perform the perception, judgment, and control required for autonomous driving, and transmits the result to the Adaptive AUTOSAR Platform through the ROS2 Bridge. Using this architecture, various robotics systems, sensor drivers, and sensor processing technologies based on ROS2 that are in development, as well as various other software currently available in the open-source community, can be implemented in the vehicle. It

is also useful for rapid prototyping because the tests required for vehicle development and the use of various sensors can be handled on the ROS2 Platform, where many drivers and sensor processing algorithms are already available and thus do not have to be developed on the Adaptive AUTOSAR Platform. There are even open sources for LiDAR and vision-related machine learning object detection and detection that are being actively researched and applied to autonomous driving, enabling more advanced autonomous driving implementations [29,30]. Since it is connected using SOME/IP, it has the potential to work with not only Adaptive AUTOSAR, but also Classic AUTOSAR, which is currently used continuously in the automotive industry for safety. This means that it can be quickly adapted to the existing automotive industry and can be implemented at a low cost because the hardware requirements are lower than other communication protocols. Furthermore, as research to integrate ROS2 into the automotive industry continues, and to meet safety standards, it can be applied to real vehicles by using the strengths of ROS2 and Adaptive AUTOSAR beyond testing and prototyping.

The limitations and future development challenges of this study can be summarized as follows.

1. The Adaptive AUTOSAR Platform used in this study is an open-source platform that partially satisfies the AUTOSAR standard and is not software that is used in the actual vehicle industry. SOME/IP, which was the focus of this study, satisfied the AUTOSAR standard and was implemented in ara::com. However, in actual vehicles, various other factors that are not implemented in this open source may cause unexpected conflicts. More testing and research are needed.

2. The validation for this study was performed via simulation only. Hardware constraints, such as computing power, should be considered when research to integrate ROS2 and the Adaptive AUTOSAR Platform for autonomous driving are integrated for real devices.

3. In this study, we did not deeply consider network topology selection or traffic optimization [31,32] and focused solely on implementing and verifying interoperability through SOME/IP between Adaptive AUTOSAR and ROS2. Further consideration should be given to more suitable QoS settings or traffic optimization in actual communication environments.

Through this study and other extended research, the automotive industry and autonomous driving technologies that are developing in different areas will be combined. Ultimately, the ROS2 Platform and the Adaptive AUTOSAR Platform will complement each other, helping to reduce the time required for testing and prototyping during development. This will lead to faster advancements in autonomous driving and hopefully create a better transportation environment, including moving away from the congestion of traditional traffic and paving the way for reducing the negative aspects of transportation, such as energy consumption and emissions [33].

**Author Contributions:** Conceptualization, D.H. and C.M.; methodology, D.H.; software, D.H.; validation, D.H.; formal analysis, D.H.; investigation, D.H.; resources, C.M.; data curation, D.H.; writing—original draft preparation, D.H.; writing—review and editing, D.H. and C.M.; visualization, D.H.; supervision, C.M.; project administration, D.H.; funding acquisition, C.M. All authors have read and agreed to the published version of the manuscript.

**Funding:** This paper was supported by the Korea Institute for Advancement of Technology (KIAT) grant funded by the Korea Government (MOTIE) (P0020536, HRD Program for Industrial Innovation).

**Data Availability Statement:** Data is available from the authors.

**Conflicts of Interest:** The authors declare no conflicts of interest.

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
