# Peer review of "Autonomous Driving System Architecture with Integrated ROS2 and Adaptive AUTOSAR"

_electronics, doi:10.3390/electronics13071303_

Round 1

Reviewer 1 Report

Comments and Suggestions for Authors

This paper introduces an architecture for interoperability between AUTOSAR and ROS2 platforms, named Autonomous Driving System with Integrated ROS2 and Adaptive AUTOSAR (ASIRA).

1. Give an in-depth analysis of the experimental results (tables and figures). 

2. In the abstract, ROS2 appears for the first time on line 11. Give its full name;

3. Authors should consider the safety, economics and environmental impact of the proposed solution;

4. More tests are needed to judge the solution's performance;

5. Consider certain performance measures of the proposed solution;

6. Consider a very simple real-world device to test the feasibility of your solution;

7. Give a short introduction between each section title and the first sub-section title. Examples: "2. Background and related work" and "2.1 Adaptive AUTOSAR"; "2.2 ROS" and "ROS 1".

8. Several figures have a poor resolution;

9. The authors talk about real-time response, but each request mobilizes several exchanges!

10. Does the proposed solution support machine learning model conversion such as environment detection/segmentation accurately and efficiently? 

11. Does your system have the capacity to cover all AV functionalities such as (* localization system, road mapping system, trajectory planning system, etc.). By this I mean does your system have the capacity to receive data from the various AV sensors and provide results for each system;

12. In line 488, give the time required for data exchange between brackets and justify the "real-time" confirmation;

13. Study the stability of the response for well-known perturbation densities of the numerical data;

Author Response

Thank you for your meticulous review and high-quality feedback.

Reviewer 2 Report

Comments and Suggestions for Authors

This paper studies how ROS2 and AUTOSAR can be integrated to provide a platform for the development of automotive applications. The introduction is easy to read and leads quite well to the topic.

However, the choice of SOME-IP as SOC protocol is not justified. The authors explain that DDS has been adopted in ROS2, however for some reason that is not explained decided to use SOME-IP in their proposal. Actually, in the conclusions, they highlight the use of SOME-IP as a potential drawback. There are several studies available on the advantages of DDS for automotive, even on the integration of DDS with TSN. In fact, the integration of DDS in newer versions of AUTOSAR has already started. See https://www.rti.com/blog/status-of-dds-in-autosar. I encourage the authors to review the literature available on this topic and better explain their choice for SOME-IP, or change to DDS. It seems to me that the ROS2 to SOME-IP bridge is not necessary if DDS is used. It is important to clarify this point because it is essential in order to evaluate the real contribution of this work.

I will deep dive in the implementation details after this has been clarified. I have no doubt that the authors have done this implementation properly, but that is not a research contribution. Hence, the importance of improving the justification of their choices.

Regarding the evaluation section, figure 9 shows some screenshots of a command line that is not readable. I would propose to improve the visual representation of whatever this is meant to explain.

Regarding the dummy data used in the experiments, how was this built? It is typically very difficult to have realistic datasets for simulations. How was this data created? How realistic is it expected to be?

Regarding references, it seems that the literature review of this work is not very exhaustive. I encourage the authors to deep dive into existing works on the same matter to better frame their research work.

Author Response

(The authors gave the same response as above.)

Reviewer 3 Report

Comments and Suggestions for Authors

To develop the next-generation 9 autonomous driving and connected cars, some platforms have been realized. The paper proposes an architecture that enables the communication between two such platforms (Adaptive AUTOSAR and ROS2).

The original part is the method for integrating two platforms via the Ethernet-based SOME/IP protocol to ensure safety while maintaining a flexible environment for the development and testing of autonomous vehicles. Research in the field of autonomous vehicles is still in its beginning. That is why specialized literature is not very developed. This paper addresses this gap.

The paper proposes a new architecture for the communication between to platforms (Adaptive AUTOSAR and ROS2).

The methodology is well described.

The conclusions are consistent with the evidence and the arguments presented.

The References need to be seriously revised. There are large inconsistencies between the References and the text. For example, I don't think that [17] and [21] in the text correspond to the ones in the References. In the text does not appear [18]. In the text appears [25] and [26], and the bibliography stops at [23].

Authors should redo the bibliography. There are many inconsistencies with the text. At [17] and at [21] I don't think it is correspondence. [18] does not appear in the text. In the text it is [25] and [26], but the bibliography is only up to [23].

Fig. 10 is unclear and confusing with multi-digit numbers.

Author Response

(The authors gave the same response as above.)

Round 2

Reviewer 1 Report

Comments and Suggestions for Authors

Dear authors,

Thank you for having taken into consideration all my comments and for having tried to answer each of them with sincerity, which has improved the quality of your paper. 

Reviewer 2 Report

Comments and Suggestions for Authors

The authors have correctly addressed my concerns. I believe the manuscript is now in good shape for publication.